# 312 MAX Phases: Elastic Properties and Lithiation

**DOI:** 10.3390/ma12244098

**Published:** 2019-12-08

**Authors:** P.P. Filippatos, M.A. Hadi, S.-R.G. Christopoulos, A. Kordatos, N. Kelaidis, M.E. Fitzpatrick, M. Vasilopoulou, A. Chroneos

**Affiliations:** 1Faculty of Engineering, Environment and Computing, Coventry University, Priory Street, Coventry CV1 5FB, UK; filippap@uni.coventry.ac.uk (P.P.F.); ap.kordatos@gmail.com (A.K.); ad1978@coventry.ac.uk (N.K.);; 2Institute of Nanoscience and Nanotechnology (INN), National Center for Scientific Research “Demokritos”, AgiaParaskevi, 15341 Athens, Greece; 3Department of Physics, University of Rajshahi, Rajshahi 6205, Bangladesh; hadipab@gmail.com; 4Department of Materials, Imperial College London, London SW7 2BP, UK

**Keywords:** MAX phases, DFT, elastics, lithiation

## Abstract

Interest in the M_n+1_AX_n_ phases (M = early transition metal; A = group 13–16 elements, and X = C or N) is driven by their ceramic and metallic properties, which make them attractive candidates for numerous applications. In the present study, we use the density functional theory to calculate the elastic properties and the incorporation of lithium atoms in the 312 MAX phases. It is shown that the energy to incorporate one Li atom in Mo_3_SiC_2_, Hf_3_AlC_2_, Zr_3_AlC_2,_ and Zr_3_SiC_2_ is particularly low, and thus, theoretically, these materials should be considered for battery applications.

## 1. Introduction

MAX phases are a class of ternary nitrides and carbides that we can categorize into families of M_(n+1)_AX_n,_ with n = 1, 2, 3, 4, and other MAX phase-related structures like the 321 MAX phases or the iMAX phases [1,2,3,4]. A family of 211 of these materials was discovered in powder form in 1960, with the name H-phases from Nowotny et al. [5], but, after many years, Barsoum and El-Taghy synthesized the first bulk MAX phase, Ti_3_SiC_2_ [6], with enough purity to enable them to characterize its properties. Since then, there interest has grown regarding MAX compounds, due to their unusual properties, which are a result of their bonding characteristics and their structures [7,8,9,10,11,12,13,14]. Just like MXenes, which are the corresponding binary carbides and nitrides that are created from MAX phases after an exfoliation process that removes the A layer [7], the MAX compounds have a high elastic stiffness and are good electrical conductors [8]. Regarding their mechanical behavior, MAX phases are machinable and have high thermal and damage resistance [11,12]. These properties have constituted the 312 MAX phases as important materials for numerous high-end applications, including space, electronic, and nuclear [13,14,15,16,17,18,19,20,21,22,23,24,25].

A schematic of the crystal structure (*P6_3_/mmc*, space group no. 194) [5] of the 312 MAX phases is given in Figure 1. The “metallic” layers (A) are positioned between the *n* “ceramic” layers (M_3_AX_2_ for *n* = 2) along the *c*-direction. The M and X layers effectively form M_2_X slabs, with face-centered cubic-type stacking. In the present research, we have examined the potential application of the 312 MAX phases as anode materials in Li-ion batteries or supercapacitors. Modern battery technology uses the 3D carbon structure, graphite, as an anode material, and there are reports that graphene batteries that could have better performances [26]. Etching the A from the MAX compounds leads to the 2D material MXenes, which, from a theoretical point, could become the future Li-ion battery anode material, as it exhibits better cycles than graphite [27,28]. However, in order to create the MXenes, the method of hydrofluoric acid (HF) etching is used, and although it removes the A layer, it also affects the bonding between the M and X layers. This creates some unwanted characteristic changes, as it sometimes affects the elastic properties of the material. Moreover, although in most cases MXenes have unique electrical characteristics, sometimes when the etching removes the A layer, the resulting structure becomes a semiconductor [29], and as a result, this MXene cannot be used as an anode material. Recently, the first 2D MAX phases (MAXenes) were demonstrated that consist of a 2D structure, which, unlike the 2D MXenes, keeps the A layer [30]. In order to examine the application of MAXenes in Li-ion batteries, it is evident that the interactions of Li in the MAX phases should be examined first in order to predict how the lithiation is affected by the A layer. Lastly, as has been indicated in previous studies, the creation of MXenes is a high-cost method [29]. From all of the above, we believe that our research is important, as the Li-doped MAX phases could have better performances in the Li-ion batteries technology than MXenes, and moreover, they have never been investigated before.

The aim of the present work is to study the elastic properties and Li formation in the M_3_AX_2_ phases (M = Hf, Nb, Ta, Ti, V, Zr, and Mo; A = Si, Al, Sn, Ga, and In; X = C) using the density functional theory (DFT), and to compare our results with other related experimental and theoretical studies. Although Mo_3_SiC_2_, Hf_3_AlC_2_, Zr_3_AlC_2,_ and Zr_3_SiC_2_ are of interest, it should be noted that not all of these compounds have been synthesized yet, although there are many reports that provide some of them in stable forms. Specifically, Zhou et al. [31] have synthesized an Zr_3_AlC_2_ together with Solvas-Zapata et al. [32], who synthesized Zr_3_(Al_1−x_Si_x_)C_2_. Regarding Mo_3_SiC_2_, there are many theoretical reports [33] about its properties, like self-healing. However, it has not been synthesized. There are also experimental reports on stable forms of Zr_3_SnC_2_ and Hf_3_AlC_2_ from Lapauw et al. [34]. There are also some trends in thin film MAX phases [35], as well as in sol-gel methods of creating the MAX phases compounds [36], that are examined by experimentalists for the synthesis of new MAX phases. To conclude, of the four 312 materials that we propose should be examined for potential lithiation, two of them have already been synthesized in stable forms (Zr_3_SnC_2_ and Hf_3_AlC_2_) and one has been synthesized as part of a mixed structure (Zr_3_SiC_2_); accordingly, we believe that their lithiation abilities could be investigated both experimentally and theoretically. We will then propose some experimental works with the stable synthesized forms of the 312 MAX phases that could be examined, in order to see their performances as Li-ion battery anodes.

## 2. Computational Methods

CASTEP, a plane-wave DFT code, was employed for all the calculations [37,38]. The generalized gradient approximation, ultra-soft pseudopotentials [39], and the Purdew, Burke, and Ernzerhof (PBE) [40] exchange-correlation function were used. To optimize the geometry, the Broyden–Fletcher–Goldfarb–Shanno (BFGS) minimizer was employed and implemented in the CASTEP. The supercells contained 108 atomic sites, with a plane-wave basis set cut-off of 450 eV, 3 × 3 × 1 Monkhorst–Pack (MP) [41]. The Li interstitial was placed at all possible sites. After an extensive search for all the possible sites, we found all the minimum energy positions of the Li interstitials. The minimum energy sites are presented in Figure 2. For the elastic properties, a unit cell was considered, with a plane-wave energy cut-off of 550 eV and with 18 × 18 × 2*k*-point mesh.

## 3. Results

### 3.1. Elastic Properties

The mechanical behavior of materials is dependent on their elastic constants. Moreover, the elastic properties of the materials are linked to the bonding characteristics, so the information that the elastic constants provide is also connected to the chemical bonds of the atoms of the solid. The MAX compounds have hexagonal crystal structures [5,42]. Therefore, the 312 MAX phases have six different elastic constants: c_11_, c_12_, c_13_, c_33_, c_44,_ and c_66_. Only the first five of them are independent, taking into account that c_66_ = (c_11_ − c_12_)/2. In order for the MAX compounds to be dynamically stable, the following conditions must be met [43]:c_11_ > 0, c_33_ > 0, c_44_ > 0, (c_11_ + c_12_) c_33_ > 2(c_13_)^2^, and (c_11_ − c_12_) > 0(1)

The calculated results for the selected 312 MAX phases have been investigated in previous studies [44,45]. In Table 1, we present the elastic properties of the 312 MAX phases. It is seen that the above conditions are met, and so the studied MAX phases are mechanically stable.

The elastic stiffness of a solid, regarding the (100) <100> strain, is calculated by the c_11_ constant. Thus, V_3_AlC_2_ is the stiffest. On the other hand, Zr_3_AlC_2_ and Ti_3_InC_2_ are the least stiff. The c_12_ elastic constant is a measure of the deformation of the material in the (110) plane along the <100> direction. Therefore, Ti_3_AlC_2_ is the most easily deformed. The c_12_ and c_13_ values indicate that when force is applied along the a- crystallographic axis, Ti_3_AlC_2_, Ti_3_InC_2_, Ti_3_GaC_2,_ and V_3_AlC_2_ are easier to shear along the b and c axes than the other MAX compounds in Table 1. Lastly, the lower value of c_33_ for Zr_3_SnC_2_ results in the conclusion that it is easiest to deform via <001> compression under uniaxial stress.

Focusing on the bulk elastic parameters, in Table 1, the bulk modulus B, Young’s modulus Y, and the shear modulus G have been calculated. It is evident that the Zr_3_SnC_2_ has the lowest value of B, and, as a consequence, it has the lowest resistance under compression. Conversely, Mo_3_SiC_2,_ which has the highest value, has the highest resistance to compression. The shear modulus G has the lowest value in Zr_3_SnC_2,_ and so this MAX phase is more prone to shape change than the others. The Young’s modulus Y, which is a measure of the stress required for deformation, has the lowest value in the Zr_3_SnC_2_ MAX phase, compared to the other MAX phases in Table 1 (also refer to [44,45,46,47,48,49,50,51,52]).

In order to gain information about the brittle or ductile failure of the MAX phases, the Pugh’s modulus (B/G) is used [53]. More analytically, when the Pugh’s modulus exceeds 1.75, the material is characterized as ductile, which means that a crack progresses slowly when plastic deformation occurs. Conversely, in brittle materials, cracks extend rapidly with little applied stress. According to the results of Table 1, all of the MAX phases studied are brittle, except for Mo_3_SiC_2_ and Ta_3_SiC_2_. Another important parameter is the anisotropy factor k_c_/k_a_ = (c_11_ + c_12_ − 2c_13_)/(c_33_ − c_13_), which indicates whether the MAX phase is more compressible along the a- or c-axis. It is obvious that Ti_3_SiC_2_, Mo_3_SiC_2_, Hf_3_SnC_2_, Hf_3_SiC_2_, V_3_AlC_2_, and Ta_3_SiC_2_ are the only MAX compounds of those studied where compression on the a-axis has almost the same value as on the c-axis.

The Poisson’s ratio is another important constant that informs us if the material is a central-force solid or a non-central-force solid [48], and also classifies the materials as brittle or ductile [54,55]. If the Poisson’s ratio is between 0.25 and 0.50, then the material is a central-force solid. Otherwise, it is a non-central-force solid. Furthermore, if the Poisson’s ratio is more than 0.26, then the solid is ductile, and if it has a lower value, it is brittle. The calculated results show that all the studied MAX phases in Table 1 are non-central-force and brittle, except for Mo_3_SiC_2_ and Ta_3_SiC_2_, which are central-force.

The elastic anisotropy, A, is an important description meaning that a body cannot develop the same strain independently of the direction in which the stress is applied. The elastic anisotropy factor indicates how the elastic properties of a solid are dependent on the direction of the stress. Additionally, the elastic anisotropy is connected with the thermal expansion and the crystal microcracks [56]. For the MAX phase systems that are hexagonal, the elastic anisotropy factor is calculated from the equation A = 4c_44_/(c_11_ + c_33_ − 2c_13_), and if A = 1, the crystal is isotropic. The results of Table 1 characterize Mo_3_SiC_2_ and Ta_3_SiC_2_ as being more elastically anisotropic than the other MAX phases, and because the value of the elastic anisotropy factor of Hf_3_SnC_2_ is almost 1, this MAX phase is elastically isotropic.

As regards the elastic properties of the MXenes, focusing on the research of Ge et al. [57] on the superconducting and high hardness of Mo_3_C_2_, it has been proved that it is a brittle material with a B/G of 2.35. We calculated that the hypothetical Mo_3_SiC_2_ is a brittle material, and we found that the B/G is slightly lower than the similar MXene, with a value of 2.11. As a result, it is seen that the Mo_3_SiC_2_ is not as brittle as Mo_3_C_2_, which makes it more difficult to crack during “diffusion”. As regards the Ti_3_C_2_, Borysiuk et al. [58] used the molecular dynamics method in order to predict the elastic properties of the Ti_n+1_C_n_ MXenes, and they calculated, in the case of Ti_3_C_2,_ a value of 502 GPa for the Young’s modulus, while Bai calculated a c_11_ constant equal to 523 GPa [59]. Compared to our results, it is evident that for every one of the Ti_3_AC_2_ MAX compounds, the Young modulus and the elastic constant c_11_ have much lower values. Focusing on the Zr_3_C_2_ MXene, Xie et al. [60] made a theoretical study on the elastic properties of that MXene, and compared to our 312 MAX phases with Zr-A-C, it is seen that only the Zr_3_SnC_2_ is less stiff. As a result, it will be softer and more easily machinable than the Zr_3_C_2_. From all of the above, it is evident that the majority of our MAX phases are performing with better elasticity characteristics and can be more easily manipulated than the similar MXenes, in order to be used as anodes in Li-ion batteries (LIBs).

### 3.2. Lithiation

The formation energy to incorporate a Li atom in the MAX phase, with ΔH for Li-intercalated systems, is defined by the following equation:(2)ΔH=E(withxLi)−E(withoutLi)−xE(Li)where *E* (with xLi) and *E* (without Li) are the energies of the system with and without Li atoms. Herein, we used one Li atom as an interstitial, and in result, x = 1. Also, *E* (Li) is the total energy of a single Li atom (here, it is 192.029 eV). In order to calculate the energy of the one Li atom, we performed a calculation for a supercell consisting of 67 Li atoms, and we performed geometry relaxation. We thus calculated the energy of the supercell, and we divided it with 67 in order to find the energy of the one atom. In Table 2, the formation energies of the lithiated 312 MAX phases are provided.

From Table 2, it is evident that the lithiation of the 312 MAX phases studied here is endothermic, which reflects instability. However, Mo_3_SiC_2_, Hf_3_AlC_2_, Zr_3_AlC_2_, and Zr_3_SiC_2_ exhibit Li formation energies less than 0.5 Ev, making the incorporation of Li in the MAX lattice feasible, and they are even lower than those of MAX materials considered in previous works [61,62,63]. This is extremely important, as MXenes are generally considered better candidates for battery applications, compared to the MAX phases. Nevertheless, the present work identifies four materials (Mo_3_SiC_2_, Hf_3_AlC_2_, Zr_3_AlC_2_, and Zr_3_SiC_2_) that are potentially important for such applications. It should be stressed that previous experimental works have only identified oxygen-doped Ti_3_SiC_2_ as having high Li-ion storage capacity, and hence, as being potentially important as an anode material for Li-ion batteries [64]. In the present study, though, we found that Ti_3_SiC_2_ has a high formation energy for lithiation (refer to Table 2). This implies that doping could further decrease the Li-intercalation formation energy of Mo_3_SiC_2_, Hf_3_AlC_2_, Zr_3_AlC_2_, and Zr_3_SiC_2_, thus making them appropriate candidates for battery applications. MAX phases present potential advantages over MXenes, exhibiting better material properties (high thermal-shock resistance, elastic stiffness, melting temperatures, and electrical and thermal conductivity). They are less-complicated structures, as they do not need functional groups to be stabilized [61]. We searched the literature about the formation energy of similar lithiated MXene structures, but we could only find a report about Ti_3_C_2_ where the formation energy for the lithiated structure was calculated at 4.40 eV [65]. It is obvious that this value is higher than our theoretical results, so the lithiation in the 312 MAX phases needs less energy than the above-mentioned MXene.

In an effort to link the elastic properties with the Li formation energies in the 312 MAX phases, we considered Figure 3. This was motivated by initial work on the Ti_3_AC_2_ (A = Sn, Si, Ge, Ga, Al, and In) MAX phases where there is a decrease of Li formation energy with respect to the C_11_ elastic constant (refer to Figure 3a). Nevertheless, when considering the whole range of the 312 MAX phases, there is no specific trend (refer to Figure 3b–d). Future theoretical work should include thermodynamic models to investigate further if the bulk properties impact the formation energies of Li in these systems [66,67,68].

## 4. Conclusions

To summarize, the mechanical behavior and the formation energy for the lithiation of the 312 MAX phases has been calculated using the density functional theory. From our calculations, it is evident that Zr_3_SnC_2_ is more prone to shape change along the b- and c-axes when stress along the a-axis is applied. Moreover, Zr_3_SnC_2_ does not need high stress in order to deform, and has low resistance to deformation under compression.

The energy to incorporate Li-ions in the 312 MAX phases is considerably high, with the exception of Mo_3_SiC_2_, Hf_3_AlC_2_, Zr_3_AlC_2_, and Zr_3_SiC_2_, for which formation energies of Li intercalation less than 0.5 eV were calculated. The Li formation energies in Mo_3_SiC_2_ and Zr_3_SiC_2_ are particularly low. However, they have not yet been synthesized. Regarding the other compounds, it could be proposed that their potential as LIBs or supercapacitor anodes should be examined.

To conclude, from the four 312 materials that we propose should be examined for potential lithiation, two of them have already been synthesized in stable forms (Zr_3_SnC_2_ and Hf_3_AlC_2_) and one has been synthesized as part of a mixed structure (Zr_3_SiC_2_), and so we believe that their lithiation ability could be investigated both experimentally and theoretically. There have been, compared to the MXenes, very few studies of the MAX phases for battery applications. Obviously, the incorporation of Li is only part of the picture, and future studies should focus on the diffusion of Li from both an experimental and theoretical viewpoint. Doping strategies should also be employed to lower the formation and migration energies of Li in the MAX phases.

## Figures and Tables

**Figure 1 materials-12-04098-f001:**
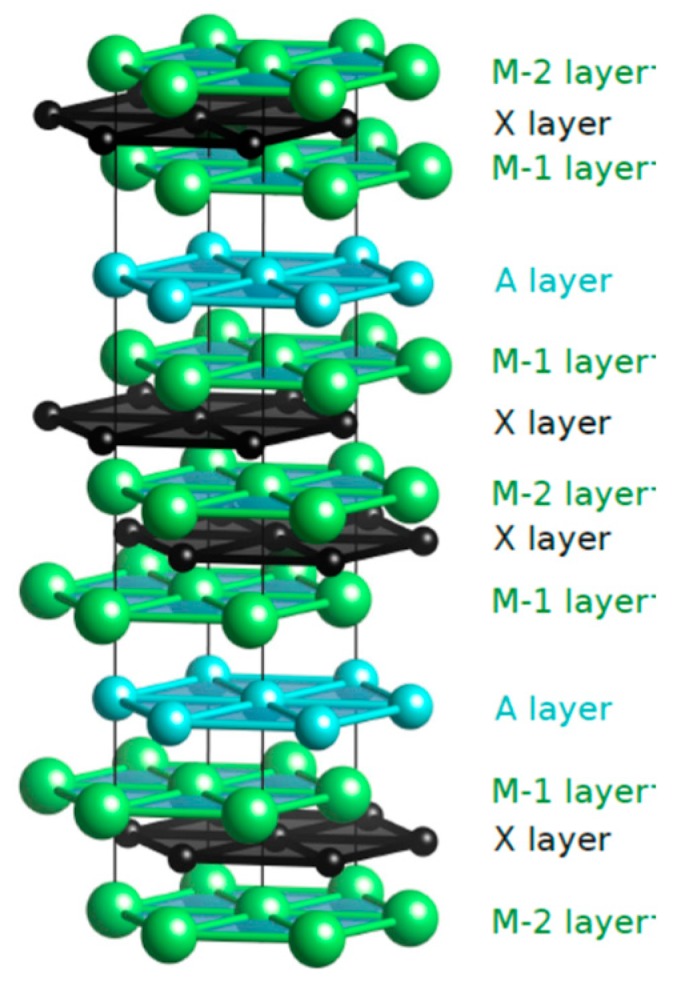
Crystal structure of the 312 MAX phase.

**Figure 2 materials-12-04098-f002:**
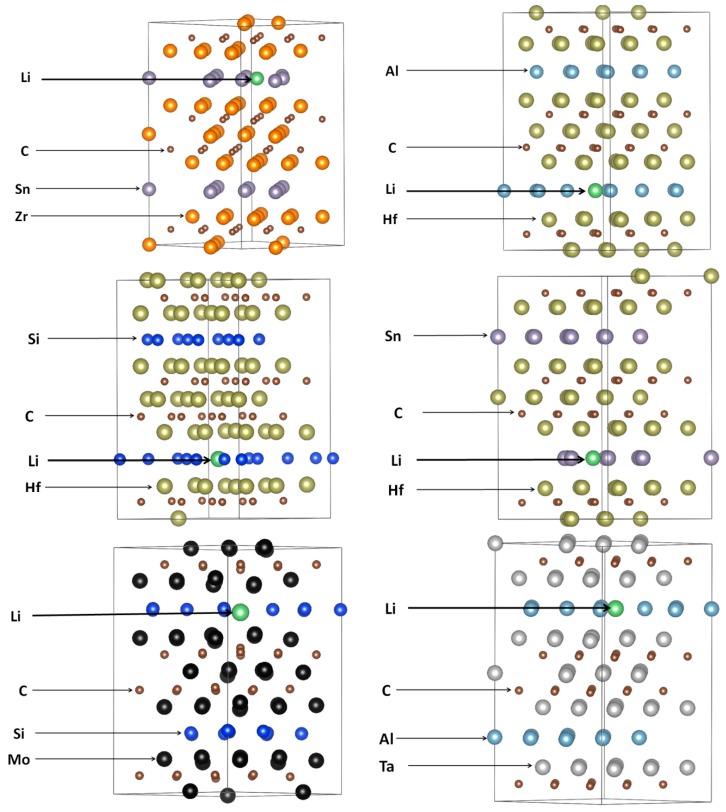
The Li interstitial positions (green atoms) in the 312 MAX phases.

**Figure 3 materials-12-04098-f003:**
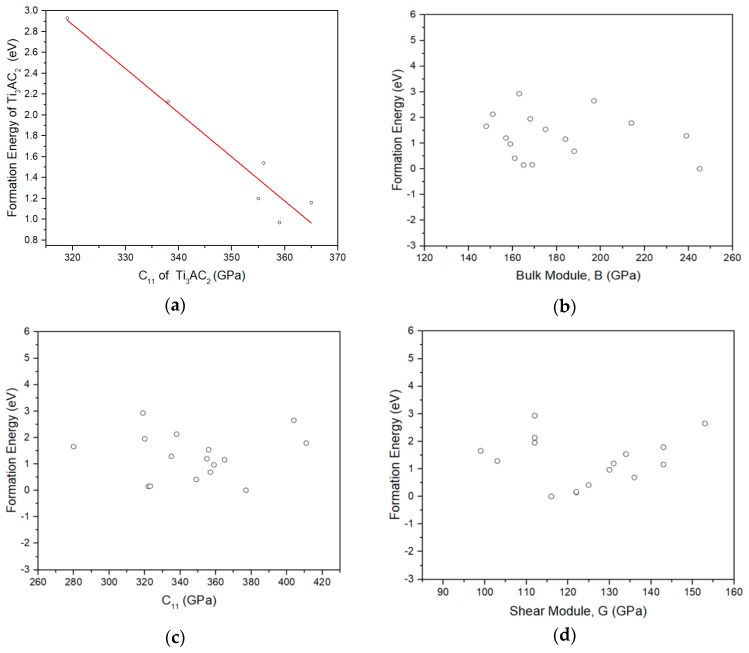
(**a**) The dependence of the Li MAX phases formation energy (eV), with respect to the c_11_ (GPa) elastic constant for the Ti_3_AC_2_ (A = Sn, Si, Ge, Ga, Al, In) MAX phases. (**b**) The dependence of the formation energy (eV) of lithiated 312 MAX phases from the bulk modulus (GPa). (**c**) The dependence of the formation energy (eV) of the lithiated 312 MAX phases from the C_11_ (GPa) elastic constant. (**d**) The dependence of the formation energy (eV) of the lithiated 312 MAX phases from the shear modulus G (GPa).

**Table 1 materials-12-04098-t001:** Calculated elastic constants Cij (GPa), bulk modulus B (GPa), shear modulus G (GPa), Young’s modulus Y (GPa), Poisson’s ratio v, Pugh’s ratio B/G, elastic anisotropy factor A, and shear anisotropy factor (kc/ka) for the 312 MAX phases *. Comparison with previous studies [44,45,46,47,48,49,50,51,52].

Phase	*c* _11_	*c* _12_	*c* _13_	*c* _33_	*c* _44_	*A*	*k*_c_/*k*_a_	*B*	*G*	*Y*	*B*/*G*	*v*	Ref.
Ti_3_AlC_2_	355	74	66	295	125	0.971	1.314	157	131	307	1.199	0.174	[44]
	358	84	75	293	122	0.974	1.343	163	127	303	1.279	0.190	[47]
	361	75	70	299	124	0.954	1.297	160	131	309	1.221	0.178	[47]
	368	81	76	313	130	0.983	1.253	168	135	320	1.245	0.183	[46]
	-	-	-	-	-	-	-	165	124	297	1.331	0.20	[49]
Zr_3_AlC_2_	322	84	97	287	138	1.330	1.116	165	122	294	1.353	0.203	This
	314	78	79	262	107	1.024	1.279	151	110	266	1.373	0.207	[46]
V_3_AlC_2_	404	84	108	361	158	1.151	1.075	197	153	364	1.288	0.191	This
	390	82	116	358	158	1.225	0.991	196	147	354	1.333	0.200	[46]
Hf_3_AlC_2_	349	79	79	283	123	0.963	1.324	161	125	298	1.288	0.192	This
	347	77	80	291	127	0.941	1.251	162	127	302	1.276	0.189	[50]
	357	82	83	283	126	0.940	1.365	166	128	305	1.297	0.193	[51]
	348	79	82	290	112	1.058	1.264	163	121	291	1.347	0.203	[52]
Ta_3_AlC_2_	411	113	136	343	156	0.772	1.217	214	143	351	1.497	0.227	This
	441	132	138	382	175	0.781	1.217	231	157	384	1.471	0.223	[46]
Ti_3_SiC_2_	365	89	99	352	156	1.202	1.012	184	143	341	1.287	0.191	[44]
	370	99	111	349	151	1.209	1.038	192	138	334	1.392	0.210	[46]
	372	88	98	353	167	1.267	1.036	185	149	352	1.245	0.183	[48]
	-	-	-	-	-	-	-	185	139	333	1.331	0.20	[49]
	-	-	-	-	-	-	-	186	144	343	1.291	0.192	[48]
Hf_3_SiC_2_	357	93	115	334	157	1.362	1.005	188	136	329	1.382	0.209	This
	348	101	120	335	144	1.300	0.972	190	127	312	1.496	0.227	[46]
Ta_3_SiC_2_	335	145	221	325	179	3.284	0.365	239	103	270	2.320	0.317	This
	352	220	210	345	182	2.628	1.126	256	102	270	2.509	0.324	[46]
Zr_3_SiC_2_	323	85	99	304	135	0.794	1.024	169	122	295	1.385	0.209	This
	320	100	107	296	125	0.804	1.090	174	113	279	-	0.233	[46]
Mo_3_SiC_2_	377	175	186	364	151	1.637	1.011	245	116	301	2.112	0.300	This
Hf_3_SnC_2_	320	95	96	300	115	1.075	1.093	168	112	275	1.500	0.227	[45]
	326	96	97	300	107	0.991	1.123	170	110	272	1.550	0.234	[46]
Ti_3_SnC_2_	319	103	80	304	113	0.976	1.170	163	112	273	1.455	0.221	[44]
	331	96	80	285	108	0.943	1.302	161	113	274	1.436	0.217	[46]
	331	91	81	299	129	1.103	1.193	162	122	285	1.328	0.208	[48]
Zr_3_SnC_2_	280	92	84	257	110	1.192	1.179	148	99	243	1.495	0.227	[45]
	297	90	87	268	95	0.972	1.177	154	98	244	1.571	0.237	[46]
Ti_3_InC_2_	338	80	63	276	92	0.754	1.371	151	111	267	1.360	0.205	[44]
	340	85	67	263	97	0.826	1.478	152	111	267	1.362	0.205	[46]
Ti_3_GaC_2_	359	78	69	292	123	0.959	1.341	159	130	306	1.223	0.179	[44]
	356	86	75	285	113	0.920	1.390	162	122	293	1.324	0.198	[46]
Ti_3_GeC_2_	356	88	91	324	140	1.125	1.125	175	134	320	1.306	0.195	[44]
	357	100	97	325	129	1.051	1.152	180	126	307	1.426	0.216	[46]
	355	85	94	338	148	1.171	1.032	177	138	312	1.283	0.207	[48]

* Elastic constants and moduli are shown in round figures.

**Table 2 materials-12-04098-t002:** The formation energy of 312 MAX phases when one Li atom is inserted.

312 MAX Phases	Formation Energy/eV
Ti_3_AlC_2_	1.1966
Zr_3_AlC_2_	0.1499
V_3_AlC_2_	2.6507
Hf_3_AlC_2_	0.4172
Ta_3_AlC_2_	1.7890
Ti_3_SiC_2_	1.1597
Hf_3_SiC_2_	0.6902
Ta_3_SiC_2_	1.2943
Mo_3_SiC_2_	0.0056
Zr_3_SiC_2_	0.1608
Hf_3_SnC_2_	1.9508
Ti_3_SnC_2_	2.9303
Zr_3_SnC_2_	1.6645
Ti_3_InC_2_	2.1320
Ti_3_GaC_2_	0.9735
Ti_3_GeC_2_	1.5416

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
