# Peer review of "312 MAX Phases: Elastic Properties and Lithiation"

_materials, 2019, doi:10.3390/ma12244098_

Round 1
Reviewer 1 Report
The authors study 312 Max phases by DFT and calculate elastic properties and lithiation (energy needed to incorporate Li). Overall, there is some interest in this paper, but it also has major shortcomings. Publication in present form is not recommended; however, perhaps these issues can be addressed in a major revision.
1. Overall comment: Generally, the paper lacks a proper context and there are some well-known problems in the modeling (see specific comments below) that are not commented on. The introduction is extremely brief and does not provide a context or motivation for the study. It should be substantially expanded, motivating the work and the specific study more exhaustively, providing better referencing to prior work on calculations of elastic properties of MAX phases. There is also a need to motivate why one should investigate lithiation at all. As the authors correctly say, most of the studied materials would not be beneficial for lithiation. MXenes would be a much better alternative for this purpose; this class of materials (2D MAX-derivatives) should be introduced and compared with to motivate why MAX phases would be interesting to study at all for this purpose.
Specific major comments:
2. The paper’s conclusion is that four 312 MAX phases are potentially interesting for lithiation. However, two of these four phases do not actually exist (Mo3SiC2 and Zr3SiC2 are not stable) and the other two are very difficult to make. This needs to be much better presented.
3. Phase stability is not included in the computational study. Thus, the paper provides a misleading picture and assumes that all the studied phases are stable. In fact, many are not. One has to account not only for intrinsic/dynamic stability (the authors does not even check that!) but also for stable competing phases. Some important points: MAX phases with A=Si are, in fact, only known in the Ti-Si-C system, so it is of little interest to study any of the other systems except for a hypothetical comparison. Another well-known example is V3AlC2, which is not stable (the combination V2AlC + V4AlC3 is much more stable in this system).
Calculating only the stability relative to the constituent elements gives, at best, a misleading picture of the stability. Accounting for the most stable competing phases will give a different picture. Typically, around 75-80% of inherently stable MAX phases (in a given system) turn out not to be stable if properly accounting for competing phases [see M. Dahlqvist, B. Alling, and J. Rosen Phys. Rev. B 81, 220102(R) etc.].
It is in itself not a concern to study hypothetical phases for comparison. However, the authors do not appear to be aware that many of the phases studied are hypothetical and present them at face value as if they existed. This point needs to be extensively reworked.
4. In the computational work, the PBE functional (regular GGA) is used. This is fine for most of the materials studied, but not for Cr-containing phases. It is well known that one needs to account for strongly correlated electrons (by an ad hoc approach such as GGA+U or by more advanced approaches such as hybrid functionals) in order to model Cr-containing materials correctly. It is also required to account for magnetic ordering of Cr and find a stable magnetic configuration. Without this, the elastic properties are also very overestimated. [see, e.g, a series of papers by Dahlqvist et al such as J Phys Cond Mat 2015 27:095601 and Journal of Applied Physics 113, 216103 (2013)].
Without this, the results on Cr-containing phases in the present paper are incorrect. The easiest is to just remove these results, they are not central to the paper.
5. How are the formation energies calculated; I cannot find this mentioned anywhere? Do the authors mean the energy for Li insertion? That is not the same thing as “formation energy” of the MAX phases (figs 3-5)
Other minor (but important comments):
6. The introduction section is generally very weak, as mentioned above. In addition to this overall point, it also contains a number of problematic or erroneous statements, and/or inaccurate use of references:
6a. “MAX phases are a class of ternary nitrides and carbides that we can categorize into families of 25 MxAyXz of xyz = 211, 312, 413 and 321 [1-3].” No, the MAX phases are defined as M(n+1)AXn, that is 211, 312, 413, and possible higher-order phases (514, etc) and intergrown phases (with alternating layers of 211/312 or 312/413). See section 2 in ref. 6 for the terminology.
6b. The “321” phases (ref. 3) are structurally related to the MAX phases, but they not MAX phases (as the authors of ref. 3 inappropriately claim). Other structurally related phases should also be mentioned, if this is brought up [see, e.g, Y. Wang, Y.C. Zhou, Annu. Rev. Mater. Res. 39 (2009) 10 or J Zhou, et al. Angewandte Chemie International Edition 55 (16), 5008 and ACS nano 11, 3841]
6c. “These materials were discovered in powder form in 1960 with the name H-Phases from Nowotny et al. [4] but after many years Barsoum synthesized the first Bulk MAX Phase, Ti3SiC2 [5].” “H phases” is a name for the 211 phases, not for the MAX phases, and Barsoum did not synthesize “the first” bulk MAX phase. Barsoum and el-Raghy made bulk Ti3SiC2 of sufficient purity to enable characterizing inherent properties, the phases had been made in bulk many times before that.
6d. “Just like MXenes, which are the corresponding binary carbides and nitrides that are created from 30 MAX Phases via etching [5],…” MXenes need to be introduced better, as mentioned above. Also, ref 5 is not the correct references for this statement.
6e. References to review articles are quite well-selected. However, references 7-8 should be replaced by Barsoum’s book [Wiley 2013] and some important updates are Sokol et al Trends in Chemistry 1, 210 2019, and Eklund et al J. Phys. D: Appl. Phys. 50 113001 2017.
There are many places in the articles where word are merged, like “phasesare”, “applicationscompared” or “Zr3SiC2thus”on p.3. Please fix everywhere in the paper.
References:
Ref.4 “H. Nowotny” not “V. H.” (“von Hans Nowotny” mean “by Hans Nowotny”, it is not an initial)
Ref. 16, here also more from the series of papers by Lapauw et al. on Zr-Al-C and Hf-Al-C MAX phases should be cited (J Eur Ceram Soc 36 1847 2016; Inorg Chem 55, 10922 (2016)
Author Response
Overall comment: Generally, the paper lacks a proper context and there are some well-known problems in the modeling (see specific comments below) that are not commented on. The introduction is extremely brief and does not provide a context or motivation for the study. It should be substantially expanded, motivating the work and the specific study more exhaustively, providing better referencing to prior work on calculations of elastic properties of MAX phases. There is also a need to motivate why one should investigate lithiation at all. As the authors correctly say, most of the studied materials would not be beneficial for lithiation. MXenes would be a much better alternative for this purpose; this class of materials (2D MAX-derivatives) should be introduced and compared with to motivate why MAX phases would be interesting to study at all for this purpose
Response: The Introduction (and Conclusions) have now been corrected and expanded. As the reviewer suggested, we focused more on the motivation and the importance of our study. We compared our results to other DFT and experimental works on MXenes and we found that in some cases the precursor MAX Phases has better properties than the similar MXene. Furthermore, we believe that as many different forms, similar to MAX Phases, are often created (for example MAXenes, 321 MAX Phases, iMAX phases etc) we should investigate first the interaction of Li atom within the MAX Phases and more specifically with the A layer which is where it is ,in most of the cases, the interstitial position as in the future these materials could exhibit an better performance than today’s anode materials.
The paper’s conclusion is that four 312 MAX phases are potentially interesting for lithiation. However, two of these four phases do not actually exist (Mo3SiC2 and Zr3SiC2 are not stable) and the other two are very difficult to make. This needs to be much better presented.
We have now discussed this extensively in the revised paper.
Phase stability is not included in the computational study. Thus, the paper provides a misleading picture and assumes that all the studied phases are stable. In fact, many are not. One has to account not only for intrinsic/dynamic stability (the authors does not even check that!) but also for stable competing phases. Some important points: MAX phases with A=Si are, in fact, only known in the Ti-Si-C system, so it is of little interest to study any of the other systems except for a hypothetical comparison. Another well-known example is V3AlC2, which is not stable (the combination V2AlC + V4AlC3 is much more stable in this system).
Response: Phase stability of the MAX phases is an interesting issue but beyond the scope of the present work, which focuses primarily on the lithiation and mechanical properties of these phases. We now highlight more the synthesized compounds are we included references 30-34. Also, we clarify that Mo3SiC2 have not been synthesized yet. It is common in the literature to find references and DFT calculations with these yet unsynthesized compounds so we have included them in the present study for comparison. Lastly, we investigated the dynamical stability with the known elastic conditions (please see reference 42) and we found that our calculated MAX Phases are all mechanically stable.
In the computational work, the PBE functional (regular GGA) is used. This is fine for most of the materials studied, but not for Cr-containing phases. It is well known that one needs to account for strongly correlated electrons (by an ad hoc approach such as GGA+U or by more advanced approaches such as hybrid functionals) in order to model Cr-containing materials correctly. It is also required to account for magnetic ordering of Cr and find a stable magnetic configuration. Without this, the elastic properties are also very overestimated . Without this, the results on Cr-containing phases in the present paper are incorrect. The easiest is to just remove these results, they are not central to the paper.
Response:
The referee is right and we have removed the Cr-containing MAX phase.
How are the formation energies calculated; I cannot find this mentioned anywhere? Do the authors mean the energy for Li insertion? That is not the same thing as “formation energy” of the MAX phases (figs 3-5)
Response: Yes as formation energy we mean the energy of the Li insertion. We corrected this in the revised text and in Figures 3-5. FIX FIGS 3-5 caption
“MAX phases are a class of ternary nitrides and carbides that we can categorize into families of 25 MxAyXz of xyz = 211, 312, 413 and 321 [1-3].” No, the MAX phases are defined as M(n+1)AXn, that is 211, 312, 413, and possible higher-order phases (514, etc) and intergrown phases (with alternating layers of 211/312 or 312/413). See section 2 in ref. 6 for the terminology.
Response: The reviewer is correct, we have made the proposed changes.
6b. The “321” phases (ref. 3) are structurally related to the MAX phases, but they not MAX phases (as the authors of ref. 3 inappropriately claim). Other structurally related phases should also be mentioned, if this is brought up
Response: As the reviewer suggested, we made the changes and we also added some references for other structures related to the MAX Phases (references 3-4, reference 30)
6c. “These materials were discovered in powder form in 1960 with the name H-Phases from Nowotny et al. [4] but after many years Barsoum synthesized the first Bulk MAX Phase, Ti3SiC2 [5].” “H phases” is a name for the 211 phases, not for the MAX phases, and Barsoum did not synthesize “the first” bulk MAX phase. Barsoum and el-Raghy made bulk Ti3SiC2 of sufficient purity to enable characterizing inherent properties, the phases had been made in bulk many times before that.
Response: As the reviewer suggested, we made the changes.
6d. “Just like MXenes, which are the corresponding binary carbides and nitrides that are created from 30 MAX Phases via etching [5],…” MXenes need to be introduced better, as mentioned above. Also, ref 5 is not the correct references for this statement.
Response: As the reviewer suggested, we made the changes and we changed the ref 5.
6e. References to review articles are quite well-selected. However, references 7-8 should be replaced by Barsoum’s book [Wiley 2013] and some important updates are Sokol et al Trends in Chemistry 1, 210 2019, and Eklund et al J. Phys. D: Appl. Phys. 50 113001 2017.
Response: As the reviewer suggested, we made the changes and we replaced the references as suggested.
Reviewer 2 Report
The article reports on a systematic investigation of the elastic properties and the incorporation of lithium atoms in 312 MAX phases using density functional theory calculations. This is a very interesting paper that ultimately should be published considering the significant importance of MAX phases for the preparation of MXenes, which are recognized as promising materials for lithium-ion batteries and supercapacitors. At present, however, the manuscript requires substantial re-working. Prior to publication, the paper should be completed and modified according to the following comments.
Comments and recommendations
Abstract
Line 19. The article reports on the incorporation of only one Li atom into MAX phases. Please, clarify!
Introduction. The introduction contains a general definition of MAX phases. However, the authors have not justified the importance of their calculations: it is not clear from the introduction on why it is important to study the elastic properties and Li formation in M3AX2 phases. This should be clarified in details. It is worth mentioning in the introduction that MAX phases have been synthesized in form of thin films. Therefore, it is worth to mention the article 10.1080/21663831.2019.1594428. Materials and methods. It is stated that “the Li interstitial was placed at all possible sites”. These positions should be clarified! Because the formation energy of MAX phases will vary depending on the site of Li atoms and depending of Li atom nearest neighbors. The formation energy will depend on the site of insertion: interstitially into MX layer or MA layer. The site of insertion should be demonstrated in Fig.1 or in separate figure. This is a critical point which should be explained in details! Results The authors well describe the difference in elastic constants, Poisson’s ratio, ductility and brittleness of MAX phases. However, the authors have not explained why this difference occur. It should be explained in terms of chemical bonding, the valence of elements, bond strength etc. in one-two paragraphs. Probably, the references on other theoretical works dedicated to MAX phases should be provided. Lines 108-110. It is not clear how this statement is related to the provided results. The calculations have been carried out for MAX phases not MXenes, right? Please, clarify! Moreover, It is not well described why Mo3SiC2, Hf3AlC2, Zr3AlC2 and Zr3SiC2 are important for batteries and for what reasons. Lines 100-101. Table 2. The equation for the calculation of the formation energy imply that the formation energy will depend from the concentration of Li in the MAX phases. However, the formation energy provided in Table 2 is provided for only one atom. Upon insertion of more than one atom, the formation energy will change. This should be explained and the formation energy for higher Li concentration should be provided. Lines 121-123. The reference on the initial work should be provided. Lines 123-124. It is very important to discuss why there is no specific relation between the elastic properties with the Li formation energies although there is a dependence of Li formation energy with respect to C11 elastic constant observed in Fig.2. Why this dependence should or should not originate? Please, clarify! I would recommend to marge Fig. 3-5 into one figure with the subscripts a), b), c). It is also not clear from the figures for which MAX phases these dependencies are plotted. Conclusions Lines 154-158. In my opinion, these sentences more fit the introduction rather than the conclusions.
The article contains very valuable calculation results, but they should be analyzed more detailed and interpreted correctly. Therefore, the major revision is necessary.
Author Response
Line 19. The article reports on the incorporation of only one Li atom into MAX phases. Please, clarify!
Response: As the referee suggested, we made clear in the text that in the present paper, we used only one Li atom. We have added more Li atoms in DFT calculations of these MAX phases (not reported in the paper) but the binding energy between Li atoms is very small.
The introduction contains a general definition of MAX phases. However, the authors have not justified the importance of their calculations: it is not clear from the introduction on why it is important to study the elastic properties and Li formation in M3AX2 phases. This should be clarified in details. It is worth mentioning in the introduction that MAX phases have been synthesized in form of thin films. Therefore, it is worth to mention the article 10.1080/21663831.2019.1594428
Response: We made the appropriate changes on the introduction of our research paper in order to justify the importance of our DFT calculations. We believe that the elastic properties and the Li formation are two of the most important tasks that should be theoretically checked in order to propose a compound as an anode material for LIBs. With the elastic properties, the shape change and the damage tolerance are checked whereas with the Li formation energy we can predict theoretically the energy “cost” in order to have a lithiation. Lastly, we mentioned the article that the referee suggested.
It is stated that “the Li interstitial was placed at all possible sites”. These positions should be clarified! Because the formation energy of MAX phases will vary depending on the site of Li atoms and depending of Li atom nearest neighbors. The formation energy will depend on the site of insertion: interstitially into MX layer or MA layer. The site of insertion should be demonstrated in Fig.1 or in separate figure. This is a critical point which should be explained in details!
Response: The referee was right to comment on the method of the exact position of the Li atom. We have examined every possible position within the MAX Phases structure. In total, we did more than 1000 calculations for all the MAX Phases in order to find the energy minimum structure. The correct interstitial position which was the one that we used in order to calculate the formation energy, is presented in Figure 2.
The authors well describe the difference in elastic constants, Poisson’s ratio, ductility and brittleness of MAX phases. However, the authors have not explained why this difference occur. It should be explained in terms of chemical bonding, the valence of elements, bond strength etc. in one-two paragraphs. Probably, the references on other theoretical works dedicated to MAX phases should be provided. Lines 108-110. It is not clear how this statement is related to the provided results. The calculations have been carried out for MAX phases not MXenes, right? Please, clarify! Moreover, It is not well described why Mo3SiC2, Hf3AlC2, Zr3AlC2 and Zr3SiC2 are important for batteries and for what reasons. Lines 100-101.
Response:
In Line 108-110 we have explained the equation of the formation energy of the lithiation with one Li atom. Moreover, as the referee suggested, we explained why these 312 MAX Phases are important for the LIBs. We were not able to explain the trends in terms of chemical bonding, the valence of elements, bond strength as requested by the referee.
Table 2. The equation for the calculation of the formation energy imply that the formation energy will depend from the concentration of Li in the MAX phases. However, the formation energy provided in Table 2 is provided for only one atom. Upon insertion of more than one atom, the formation energy will change. This should be explained and the formation energy for higher Li concentration should be provided. Lines 121-123. The reference on the initial work should be provided. Lines 123-124. It is very important to discuss why there is no specific relation between the elastic properties with the Li formation energies although there is a dependence of Li formation energy with respect to C11 elastic constant observed in Fig.2. Why this dependence should or should not originate? Please, clarify! I would recommend to marge Fig. 3-5 into one figure with the subscripts a), b), c). It is also not clear from the figures for which MAX phases these dependencies are plotted. Conclusions Lines 154-158. In my opinion, these sentences more fit the introduction rather than the conclusions.
Response: We have made some calculations on the formation energy of two inserted Li atoms in some of the 312 MAX Phases. Apart from that, we have also done the Density of States plots in order to see if there are any changes with the second inserted Li atom. We saw that in the DOS, there are no changes from the insertion of the second Li atom. As regards the formation energy we have calculated that although the formation energies have a higher value, still there are no significant changes. As the reviewer proposed, we combined Figures 3-6 to one Figure. In Figure 3a we plotted the formation energy with C11 elastic constants for only the Ti3AC2 MAX Phases. In all the other cases the plots are for all the 312 MAX Phases that we studied here.
Reviewer 3 Report
The topic of this paper is interesting and DFT is a very powerfull technique.
Introduction is very short and the aim of the study is not very clear. Indeed, Mxenes structures are mentionned but not treated here.
Authors should decribe the potential applications of lithiation.
Considering the results it is not clearly exposed how the formation energy has been obtained (is it the minimal energy found for all the configurations, or an average value?).
How E(Li) has been calculated ?
I think this paper contain very intersting results but should be re-writted to improve the global qualtiy.
Moreoever, many typographic mistakes are present in the document.
Author Response
Introduction is very short and the aim of the study is not very clear. Indeed, Mxenes structures are mentionned but not treated here.
Response: We have now expanded the Introduction considerably.
Authors should decribe the potential applications of lithiation
Response: We added the potential applications of Lithiation which is in LIBs and in supercapacitors.
Considering the results it is not clearly exposed how the formation energy has been obtained (is it the minimal energy found for all the configurations, or an average value?).
Response: The formation energy was calculated using the minimum energy structures.
How E(Li) has been calculated ?
Response: In order to calculate the Energy of the one Li atom, we performed a calculation for a supercell consisting of 67 Li atoms and we performed Geometry relaxation. Afterwards we found the energy of the supercell and we divided it with 67 in order to find the energy of the one atom
Round 2
Reviewer 2 Report
The authors have improved the manuscript considerably. However, the are some minor points needed for a correction:
1. Fig.2. Li atom should be also marked by the arrow.
2. It is not convenient to place references in the conclusions! Re-write. The red text, in my opinion, fits more with the introduction.
Author Response
We thank the referee for the assessment and the further 2 useful comments.
Regarding these comments:
1) As requested we have now marked Li atoms with arrows
2) As requested we moved the red text to the Introduction. There are no references now in the conclusions.